# Carbonized Solid Fuel Production from Polylactic Acid and Paper Waste Due to Torrefaction

**DOI:** 10.3390/ma14227051

**Published:** 2021-11-20

**Authors:** Kacper Świechowski, Christian Zafiu, Andrzej Białowiec

**Affiliations:** 1Department of Applied Bioeconomy, Wrocław University of Environmental and Life Sciences, 37a Chełmońskiego Str., 51-630 Wrocław, Poland; andrzej.bialowiec@upwr.edu.pl; 2Department of Water, Atmosphere and Environment, Institute of Waste Management, University of Natural Resources and Life Sciences, Muthgasse 107, 1190 Wien, Austria; christian.zafiu@boku.ac.at

**Keywords:** torrefaction, solid fuel, waste to carbon, circular economy, biodegradable materials, calorific value

## Abstract

The quantity of biodegradable plastics is increasing steadily and taking a larger share in the residual waste stream. As the calorific value of biodegradable plastic is almost two-fold lower than that of conventional ones, its increasing quantity decreases the overall calorific value of municipal solid waste and refuse-derived fuel which is used as feedstock for cement and incineration plants. For that reason, in this work, the torrefaction of biodegradable waste, polylactic acid (PLA), and paper was performed for carbonized solid fuel (CSF) production. In this work, we determined the process yields, fuel properties, process kinetics, theoretical energy, and mass balance. We show that the calorific value of PLA cannot be improved by torrefaction, and that the process cannot be self-sufficient, while the calorific value of paper can be improved up to 10% by the same process. Moreover, the thermogravimetric analysis revealed that PLA decomposes in one stage at ~290–400 °C with a maximum peak at 367 °C, following a 0.42 reaction order with the activation energy of 160.05 kJ·(mol·K)^−1^.

## 1. Introduction

### 1.1. Background of Current Situation

The negative impact of plastic waste accumulated in the environment (in oceans, soils, and air), including the form of microplastics, is undeniable. Most of the commonly used polymers are based on fossil resources and resistant to biodegradation, which means that once released to the environment, they will persist for a long time. Currently, there is a risk of the release of chemicals from all plastic that is unproperly landfilled into the soil and groundwater. Plastic waste that has leaked into oceans is a cause of death of marine life and is a source of microplastic that pollutes the air we breathe and water we drink [1,2,3,4].

Geyer et al. [5] estimated in 2017 that, since the 1950s, over 8300 Mt of plastics were ever produced globally, out of which 56% (ca. 4700 Mt) of the ever-produced plastics were landfilled or ended up in the environment [5]. In 2019 alone, 368 Mt of plastic were produced [6], and it is estimated that that annual production will increase by four times in 2050 [7]. To date, to cover plastic production, around 4% of the total extracted fossil fuels (e.g., natural gas, oil, and coal), are needed annually, and by 2050 this number could increase to 20% [8]. Currently, the largest plastic producers are China (31%), North America (19%), and the European Union (EU) (16%) [6]. According to “Global Plastic Flow 2018” that was prepared by Conversio Market & Strategy GmbH [9], the global plastic consumption was 385 Mt which consisted of 172 Mt of packaging waste and 213 Mt of non-packaging waste. At the same time, 250 Mt of plastic waste was generated, of which only 175 Mt was collected, and hence 75 Mt was improperly disposed or released to the environment. Only ~28.5% of the collected plastic waste was recycled; a similar amount was incinerated, and 43% was landfilled [9].

A large share of plastic materials (almost 40% in the EU) is used for packaging, which has the shortest life cycle. Other sectors that consume large amounts of plastic are building and construction (~20%) and automotive (~10%) [6]. These shares are most probably similar for the rest of the world. According to the Ellen Macarthur Foundation [10], only 14% of produced packaging plastic globally was collected for recycling purposes, wherein 4% was lost during recycling processes, 8% was recycled in cascaded recycling (waste plastic was converted into other, lower-value products), and only 2% of produced plastic had a closed-recycling loop (wasted plastic was converted into the same or similar quality products) [10].

At the first glance, the presented data show that the most abundant type of plastic waste (packaging) is hard to recycle, or its recycling is not economical yet. The reasons for this are the low quality of the recycled plastics in comparison to the virgin material, cost-intensive recycling processes, and lack of proper infrastructure [1]. Some plastic materials, such as high-density polyethylene (HD-PE), or polyethylene terephthalate (PET) can be recycled economically, due to their high market value, whereas low-density polyethylene (LD-PE), and other foil materials are used for refuse-derived fuel (RDF) production [1,11]

### 1.2. The Problem of Bioplastic Solution

With increasing awareness of citizens about ecology and sustainability, an increasing number of producers replace conventional packing plastic with biobased and biodegradable plastics. In 2020, around 47% of all produced bioplastic was used in the packaging sector. According to the European Bioplastics organization, bioplastic (biodegradable and non-biodegradable) represents about 1% of all produced plastic. The organization also estimates that, due to the rising demand, the bioplastic market will increase by ~40% up to 2025 [11,12].

Bio-based plastics are a potential solution for problems related to fossil-based plastic. In theory, bioplastics open new end-of-life scenarios, such as composting or anaerobic digestion, and lead to a reduction in conventional plastics pollution. In practice, however, there are problems with proper management [11]. Different biodegradable plastics need different environmental conditions to be biodegraded, e.g., biodegradable PLA-based biowaste bags need relatively high temperatures for overcoming the glass transition temperature (~70 °C) and initiating biodegradability. Such temperatures can be achieved in industrial composting plants, but not in home composters. In practice, biodegradable plastic is not usually decomposed during anaerobic digestion [13]. As a result, some countries, and some municipalities in the EU, allow the use of biodegradable bags for kitchen waste collection, while others do not [13]. At the same time, the bioplastic products also increase their share in the municipal solid waste (MSW) stream as, to date, no strategies exist for the collection and processing of bioplastic wastes. The reason for this is that these plastics are still a minority in the waste stream, are difficult to detect, and require sophisticated methods for proper separation [14]. Therefore, most of the bioplastic waste goes to residual fraction of municipal solid waste or is collected with conventional plastic. In both cases, biodegradable plastics are used for RDF production or are landfilled, if the local regulations allow it. As a result, biodegradable plastics do not lead to a decrease in plastic pollutions and additionally decrease the calorific value of RDF made from waste. The calorific value of the most abundant plastic (PE-LD) used for RDF production is ca. 40 MJ∙kg^−1^ [15], while the most common biodegradable plastic used to replace it, is PLA with ~19 MJ∙kg^−1^. A simple simulation in Figure A1 shows that when biodegradable plastic share increases, the high heating value of RDF decreases from 28 to 18 MJ∙kg^−1^.

### 1.3. The RDF Quality Importance

Refused-derived fuel (RDF), also known as solid recovered fuel (SRF), is mainly made from MSW. The RDF can also be made from other waste such as used tires, sewage sludges, textiles, wood, and others. The main properties of RDF decisive of its quality are calorific value and ash content. The higher the calorific value and lower ash content, the better quality of RDF. The calorific value of RDF depends on the share of RDF components and can differ from 11 MJ∙kg^−1^ [16] to 36 MJ∙kg^−1^ [17]. From a calorific point of view, the most valuable materials are plastics such as PP and PE ~46 MJ∙kg^−1^, PS ~41 MJ∙kg^−1^, and PET ~26 MJ∙kg^−1^ [18], whereas organic waste, paper, and fabrics lead to a decrease in RDF energetic potential [19]; however, this increases the renewable energy availability. Organic waste such as kitchen and food wastes are also the main source of moisture that further decreases the energetic potential of RDF [19]. Similarly, the ash content of RDF depends on materials share, and the ash amount in plastic wastes is much lower than in other waste.

The high-quality RDF is needed for specialized incineration plants and for cement plants where RDF replaces coal and provides cleaner and partly renewable energy. In particular, cement plants need high calorific value RDF to keep the cement production process stable and safe for the environment. During waste incineration (also applies to RDF), there is a need to keep the temperature of exhaust gases above 850 °C for at least 2 s to eliminate the formation of harmful compounds. In the case of a cement plant, the waste needs to generate higher temperatures for clinker burning, and when the RDF calorific value is not high enough, the required temperature will not be obtained [20].

The RDF is usually produced in the mechanical-biological treatment plant (MBT), where MSW are valorized by various mechanical and biological methods. Mechanical methods include material separation, screening, and grinding [21]. These methods are applied to increase calorific value, increase homogeneity, and decrease ash and other pollutants (Hg, Cl) content. On the other hand, a biological method such as bio-drying is used to remove water from MSW. If RDF, produced in the MBT plant, does not meet the required quality, it can be upgraded in the future by mixing other more energetic industrial materials, by the densification process (pelletization), or by thermal processing such as torrefaction or carbonization in low temperatures [21,22]. Thermal processing in conventional pyrolysis temperatures is not applicable, as most plastics are concerting into oil and gas instead of solid carbonized fuel; as result, the calorific value of solid carbonized fuel starts to decrease [23]. Furthermore, mixing, densification, and thermal processing can be combined to maximize the quality of RDF. Here, it is important to note that each of the mentioned processes requires energy, and the legitimacy of the use of these methods depends on a specific situation.

While conventional plastics PP, PE, and PET are usually subjected to mechanical recycling after separate collection, biodegradable plastics recycling has not been developed yet. Therefore, the decreasing share of conventional plastics and increasing share of bioplastics in RDF induces a need for research on the torrefaction of these biodegradable materials, as a perspective for CSF production from MSW in the future.

### 1.4. Study Aim

In this work, PLA wastes, PLA-made cups, and paper-made cups with the addition of PLA were subjected to thermal processing-torrefaction. The main aim was to check the legitimacy of low-temperature processing of PLA wastes for fuel parameters improvement. PLA wastes were processed at 200–300 °C to check the possibilities of thermal upgrading. As torrefaction and low-temperature pyrolysis of mixed waste turned out to increase the calorific value of RDF [23], we assumed that similar results will be obtained for PLA wastes. As result, a decreasing calorific value of MSW and RDF with an increasing biodegradable plastic share will be overcome. For this reason, the fuel properties of torrefied PLA wastes, torrefaction kinetics, and theoretical energy required for torrefaction were determined.

### 1.5. Methods of Thermal Processes Analysis

There are many various methods and techniques for thermal study performance and thermal process analysis. The most common are studies using small, lab-scale reactors made for specific situations, or by adopting other equipment such as muffle furnaces or autoclaves. These types of equipment allow performing thermal conversion of materials to produce enough carbonized material used for other analyses such as proximate analysis, elemental analysis, etc. Such small reactors are in favor of testing new and non-standard materials as they provide a lot of information about process efficiency and product quality [24]. On the other hand, these reactors have limited potential for thermal process reaction analysis. Most of them work similar to a black-box and only the beginning and the final product is measured, without intermediates. For that reason, thermal analysis is also performed using thermogravimetric equipment, allowing us to measure changes in materials mass, occurred reactions, quality, and chemical compositions of intermediate products. The basic thermogravimetric analysis is TGA that provides information about mass losses during a time at a defined temperature, and differential scanning calorimetry (DSC) provides information about energy flow through sample. Additionally, TGA/DSC equipment can be coupled with other instruments that identify released gasses and their chemical composition. As result, emissions and evolved pollution during the process can be quantified and managed [25,26,27].

## 2. Materials and Methods

### 2.1. Materials

The samples of biodegradable materials for the experiment were prepared from commercially available one-use cups. Paper (PAP) served as reference material and was obtained from cups that were made of 99% of paper, and 1% of PLA. The PLA material was obtained from cups made of 100% PLA plastic. The paper cups were ground using a laboratory knife mill (Testchem, model LMN-100, Pszów, Poland), through a 3 mm sieve, while the PLA cups were cut manually into pieces of ~1 cm^2^ as the PLA was melting and blocked the mill. Then, the crumbled material was subjected to a torrefaction process. Samples of raw and torrefied materials were stored in plastic containers at room temperature (~20 °C).

### 2.2. Methods

Before the experiment, raw, crumbled materials were dried at 105 °C using a laboratory dryer (WAMED, KBC-65W, Warsaw, Poland) until a constant mass was obtained. These dry materials were used for CSF production. After that, the materials and produced CSFs were subjected to proximate analysis and higher heating value (HHV) determination analysis. Next, dry raw samples of raw materials were subjected to thermogravimetric analysis (TGA) for kinetic parameters determination and differential scanning calorimetry analysis (DSC) for determination of endo and exothermal reaction presence. Next, data from the CSF production process and proximate analysis were used to build regression models that show and describe quantitatively the effect of process temperature and time on CSF properties. 

#### 2.2.1. Torrefaction Process—CSF Production

The CSF was produced at different temperatures of 200–300 °C in intervals of 20 °C and kept for 20, 40, and 60 minutes each. For the torrefaction procedure, 10 g of dry samples were placed in ceramic crucibles. These crucibles were placed into the chamber of the muffle furnace (Snol 8.1/1100, Utena, Lithuania), which was purged with CO_2_ gas to create an inert atmosphere before the samples were heated to the setpoint temperature. During the torrefaction process, CO_2_ gas was continuously supplied to the chamber to prevent sample ignition. The CO_2_ flow was shut off after the treatment period and when the temperature of the chamber declined to <150 °C. The mass of samples before and after the process was used to calculate mass yield following Equation (1). Then, using the results of HHV, the energy densification ratio was calculated (Equation (2)), and then the energy yield of CSF was determined according to Equation (3).
(1)$$MY=\frac{m_{b}}{m_{r}}\cdot 100\;$$
where MY. is the mass yield, %; $m_{b}$ is the mass of material after torrefaction, g (CSF); and $m_{r}$ is the mass of material before torrefaction, g.
(2)$$EDr=\frac{HHV_{b}}{HHV_{r}}$$

Where EDr. is the energy densification ratio; $HHV_{b}$. is the high heating value of material after torrefaction (J·g^−1^) (CSF); and $m_{r}$ is the high heating value of material before torrefaction (J·g^−1^).
(3)EY=MY·EDr
where EY is the energy yield, %; MY. is the mass yield, %; and EDr is the energy densification ratio.

#### 2.2.2. Proximate Analysis and HHV Determination

For all samples, the proximate analysis was performed. The moisture content (MC) was determined by the drying method at 105 °C using a laboratory dryer (WAMED, KBC-65W, Warsaw, Poland) according to PN-EN 14346:2011 standard [28]. The volatile matter (VM) was measured by a thermogravimetric method using a tubular furnace (Czylok, RST 40 × 200/100, Jastrzębie-Zdrój, Poland), according to [29]. The ash content (AC) was measured by sample incineration in a muffle furnace (Snol 8.1/1100, Utena, Lithuania) according to PN-Z-15008-04:1993 standard [30], and fixed carbon was measured by difference. Additionally, samples were tested for volatile solids content (VS) and combustible part content (CP) using the muffle furnace (Snol 8.1/1100, Utena, Lithuania) according to PN-EN 15169:2011 [31] and PN-Z-15008-04:1993 [30] standards, respectively. All samples were tested for high heating value using a calorimeter (IKA, C200, Staufen, Germany), according to PN EN ISO 18125:2017-07 [32]. To ensure repeatability, each experiment was triplicated.

#### 2.2.3. Statistical Analyses 

Results of CSF production and proximate analysis were subjected to regression analyses to provide empirical equations. These equations are used to describe the following properties of CSF: MY, EDr, EY, VM, AC, FC, VS, CP, and HHV depending on process temperature and time. The regression was performed according to previous work [19]. In brief, experimental data were subjected to four regression models: (I) linear equation, (II) second-order polynomial equation, (III) factorial regression equation, and (IV) response surface regression equation. Then, determination coefficient (R^2^) and Akaike value (AIC) were calculated for each model. Next, models with the greatest R^2^ and the lowest AIC value were chosen as the best fit to experimental data; the other models were rejected. In the case chosen model had some insignificant regression coefficients (a_n_), they were removed, and regression analysis was performed again.

To check if process conditions have an impact on fuel properties, ANOVA was performed, with a post hoc Tukey test to test the pairwise significance (*p* < 0.05). 

#### 2.2.4. Thermal Analysis 

The dry samples were subjected to TG/DTG/DSC thermal analysis using a simultaneous thermal analyzer (Netzsch, 449 F1 Jupiter, Selb, Germany). Term TG/DTG/DSC stands for thermogravimetry/difference thermogravimetry/differential scanning calorimetry. TG/DTG results present how material decomposes in the function of temperature, while the DSC results show transformations and reactions occurring at a particular temperature. 

The sample was placed into a corundum crucible. The mixture of nitrogen and argon 4:1 was used as an inert gas. The sample was heated 10 °C∙min^−1^ from 30–800 °C. As a reference, an empty crucible was used. TGA/DTG/DSC analyzer automatically recalculated DSC data to mW·mg^−1^ and determined DTG from TG.

The TG data was used to determine kinetic parameters according to the Coats–Redfern (CR) method. The CR’s kinetic triplet is activation energy (Ea), pre-exponential factor (A), and order of reaction (n). The methodology of CR determination was presented elsewhere [24].

#### 2.2.5. Theoretical Mass and Energy Balance of the Torrefaction Process 

Using part of the data from analyses that have been mentioned in the earlier paragraphs, theoretical energy balance for the torrefaction of PLA and paper waste was calculated. The calculations refer to the production of 1 g of CSF and include the determination of the:Mass of substrate used to produce 1 g of CSF;Energy contained in the raw material used to produce 1 g of CSF;External energy provided to the reactor to heat the proper amount of substrate to setup temperature, to produce 1 g of CSF;Energy contained in 1 g of CSF;Mass of gas generated during the production of 1 g of CSF;Energy contained in gas after production of 1 g of CSF.

For calculations, data of MY, HHV, and DSC results were used. The scheme of energy balance determination is shown in Figure 1. The green squares represent the order of calculations, the grey squares represent experimental/calculated data used for energy balance determination, and the blue squares stand for input and output data results.

In step I, the mass yield of CSF production was used to determine the mass of substrate to produce 1 g of CSF by Equation (4), which allowed us to calculate the energy contained in the substrate used to produce 1 g of CSF by Equation (5).
(4)$$M_{s}=\frac{Mr_{CSF}}{MY_{CSF}}$$
where: $M_{s}$—mass of substrate used to produce the required amount of CSF, (here 1 g), g; $Mr_{CSF}$—required mass of CSF, (here 1 g), g; and $MY_{CSF}$—mass yield of CSF production, % (Equation (1)).
(5)$$E_{s}=M_{s}\cdot HHV_{s}$$
where: $E_{s}$—energy contained in the substrate used to produce CSF, J; $M_{s}$—mass of substrate used to produce CSF, g; and $HHV_{s}$—high heating value of substrate, J·g^−1^.

For step II, the results from DSC were used as input in the form of a power flow by the sample during heating. The DSC was converted from mW·mg^−1^ to J mg^−1^ by the multiplication by time in seconds, providing information about the energy in J used to increase the temperature for 1 g of substrate. The energy demand to heat to setpoint temperature and mass of substrate demand produce CSF per g were used to calculate the demand of external energy to produce 1 g of CSF.

For step III, it is assumed, that the energy contained in 1 g of CSF equals the HHV, which was determined by the experiment.

In step IV, the energy contained in the gas was calculated indirectly. The energy in the gas is assumed to be a sum of external energy from step II, and the difference between energy contained in substrate and energy contained in CSF obtained from torrefaction, following Equation (6).
(6)$$E_{gas}=E_{external}+E_{substrate}-E_{CSF}$$
where:$\;E_{gas}$—energy contained in the gas, J; $E_{external}$—external energy provided to the-reactor to heat the substrate to setup temperature, J; $E_{substrate}$—energy contained in the substrate used to produce CSF, J; and $E_{CSF}$—energy contained in produced CSF, J.

To keep calculations as simple as possible, the calculations were performed following assumptions:Moisture content in substrate = 0%;External energy is used to provide heat for the process;No heat losses of the reactor;The energy contained in the gas is a sum of chemical energy related to the chemical composition of gas and heat; here it was assumed that CSF is cooled down after the process, and all heat goes to gas.

## 3. Results and Discussion

### 3.1. Torrefaction Process—CSF Production

In Figure 2, Figure 3 and Figure 4, process temperature and time effect on mass yield, energy densification ratio, and energy yield of carbonized solid fuel made from PLA and PAP were presented. The equations for these models were summarized in Table A1.

The mass yield of CSF made from PLA was almost not affected by process conditions. Small weight loss was observed in CSF produced at 300 °C in 60 minutes, where the MY decrease to 92%. For comparison, MY of CSF started to decrease from the lowest temperatures, at 200 °C and 20 min, the MY had around 80%, which decreased to 40% at 300 °C and 60 min (Figure 2). The reason for the very high MY of CSF made from PLA is the PLA decomposition resistances in the torrefaction temperatures range. It has been confirmed later in this work by TG/DTG results, that PLA decomposition began around 290 °C, and peaked at 367 °C (Figure 5a). For comparison, the PAP’s main decomposition started already around 240 °C and peaked at 326 °C (Figure 5a). Although TG/DTG results are useful to investigate the thermochemical characteristics of a material, such as the temperature of decomposition, it is insufficient to determine the mass yield in certain temperature regimes or reaction times for different reactors due to different geometries, sample sizes, or thermal properties. Depending on the temperature regime, which has the main effect on decomposition, the time can result in less or more significant mass losses, especially in temperature regimes that include the main decomposition reactions and long residence time [33]. Therefore, empirical models for MY of PLA and PAP samples were developed (Table A1) to correct the challenges of the experiments.

Figure 2 shows the process temperature and time effects on the energy densification ratio (EDr). The EDr shows how much more energy is contained in the CSF in comparison to unprocessed material. When EDr is equal to 1, no effect of a process for energy improvement is observed. When EDr is lower than 1, it means that there is less energy in CSF than it was initially in a substrate, and when EDr is higher than 1, it means that there is more energy in CSF than it was in a substrate. In this study, no statistically significant (*p* > 0.05) effect of torrefaction on EDr of PLA could be observed. However, a small effect of CSF made from paper could be observed. Here, EDr increased at a statistically significant level (*p* < 0.05) at setpoint temperatures higher than 280 °C.

The studied material was characterized by low enhancement in EDr. Typically, processed biomass is characterized by EDr from 1.2 to 1.4 [34]. The EDr increase was a result of the increase in HHV. The calorific value increase was probably a result of higher deoxygenation in comparison to the less intense decarbonization of material. When torrefaction temperature increases, the relative oxygen content decreases, in favor of relative carbon content which leads to an increase in HHV of CSF [35]. In the case of PLA, the process was below decomposed temperature so proper deoxygenation could not take place, while the PAP probably did not release enough oxygen compared to carbon to significantly increase HHV. 

The energy yield (EY) shows how much energy that is contained in the substrate remains in the CSF after the process. With the increasing process temperature and time, the solid mass of substrate decreases as more gases and later also liquids are formed. Each of the products needs some chemical energy for its formation, which results in a decrease in the EY of CSF. Therefore, carbon and oxygen migration is an important factor during torrefaction [35]. The EY of CSF made from PLA was not affected by the process conditions for experimental conditions that were lower than 300 °C and 40 min (Figure 4). Under these conditions, MY remained constant and at lower temperatures, no significant changes in the HHV of torrefied PLA could be found. Therefore, the trend for EY was similar to MY. In the case of PAP, an EY decrease at temperatures higher than 280 °C was found, which resulted in a carbon migration to gas and liquid products [35,36].

### 3.2. Proximate Analysis and HHV Results

The samples of materials used to produce CSF were also analyzed for volatile matter (VM), ash content (AC), fixed carbon (FC), volatile solids (VS), combustibles parts (CP), and high heating value (HHV). The PLA materials had 100%, 0%, 0%, 100%, and 100% of VM, AC, FC, VS, and CP, respectively, while the PAP material had 88.2%, 3.6%, 8.2%, 96.3%, and 96.4% of VM, AC, FC, VS, and CP, respectively. The HHV of PLA and PAP were 19,420 and 17,525 J·g^−^^1^, respectively (Table 1). 

For PLA samples, an unexpected result was found for FC and AC, 0%, while VM, VS, and CP were 100%. The same results were obtained for all CSF made from PLA (Table 1). Moreover, the Tukey test shows that there were no significant changes between the HHV of CSF made from PLA. Therefore, it can be stated that torrefaction does not affect PLA fuel properties. These unexpected results can be explained in two ways: (I) The amount of ash (minerals) in PLA was too small to be detected by equipment that was used, or (II) there were no minerals in the PLA material at all. In case of a lack of minerals (case II), the results would be correct, as all organic matter was incinerated/devoltalized during experiments. In the other case (I), a correction for the undetected mass would have to be performed. However, the error of the undetected mass is ±0.1 mg at an input of 1 g and therefore negligible. 

In the literature, both cases can be found for PLA. In favor of assumption (II) were results from Camacho-Muñoz et al. [37] that showed 100% of vs. in a PLA sample. However, Jing et al. [38] showed that PLA is a type of thermally degradable material that burns at a relatively rapid heat release rate with negligible chars, suggesting that at least some FC should remain.

For CSF made of PAP, a decrease in VM with increasing temperature and time was observed. With increasing process temperature and time from 200 °C and 20 min to 300 °C and 60 min, the VM decreases from 86.6% to 55.7%, while FC and AC increase from 9.9% to 34.6%, and from 3.4% to 9.7%, respectively (Table 1). The observed decrease in VM is related to the devoltalization of materials. On a molecular level, large cellulose molecules in PAP are broken into smaller ones until they are small enough to be removed by convection [39]. Depending on the chemical composition, more or fewer of such small molecules are released and, as a result, different values of VM can be observed. Unlike VM, the AC and FC content increase mainly as a result of the loss in VM. Unlike AC, which is related to the mineral present in the sample, additional FC can be produced during secondary reactions [40]. Nevertheless, for biomass, the presence of components such as hemicellulose and cellulose is the main contributor of VM production, while lignin is the same for FC production [41].

Both tested materials were characterized by a relatively high level of VM, and low and zero content of FC (PAP and PLA, respectively). For comparison, wood biomass has 86% of VM, 15% of FC, and 0.4% of AC [42], torrefied wood at 300 °C in 30 min has 71% of VM, 29% of FC, and 0.4% of AC [43], while high-rank bituminous (coal) has 27.6% of VM, 65% of FC, and 7.4% of AC [44]. It is clear that fuel properties of torrefied paper and biodegradable plastic are not close to conventional solid fuels. Nevertheless, the positive aspect of PLA material is its zero-ash content, which decreases the costs for managing the ash.

The high heating value of 19.4 MJ·kg^−1^ for PLA is more than twice lower than that of conventional plastics such as polyethylene [45]. Moreover, torrefaction does not increase the HHV of PLA (Table 1). On the other hand, torrefaction was found to be suitable for PAP. The HHV of PAP increased from 17.5 MJ·kg^−1^ to 19.5 MJ·kg^−1^ in CSF produced at 300 °C; 40 min. Though these values seem to suffice when they are compared to energetic biomasses (HHV ~ 18 MJ·kg^−1^) [46], they are still small in comparison with coals 30 MJ·kg^−1^ [47] or conventional plastics 40 MJ·kg^−1^ [45].

### 3.3. Thermal Analysis Results

Figure 5a shows the TG/DTG results. The PLA mass was almost constant up to around 290 °C, where thermal decomposition started. The PLA decomposed totally in one step at temperatures of ~300–400 °C, with the maximum peak at 367 °C. Backes et al. [48] show that PLA composition (additive presence) affects thermal degradation, and some components reduce the activation energy of initiation of thermo-degradation reactions. As a result, the decomposition onset temperature and maximum peak can differ up to 40 °C depending on the processed PLA [48]. Additionally, maximum decomposition peaks occur at 353–385 °C [48]. The DSC analysis results are shown in Figure 5b. The analysis shows that during PLA pyrolysis several reactions related to polymer phase transition occurred. The first phase transition at 64 °C is the glass transition of PLA. At 149 °C, the endothermal melting transformation was observed and finally, at 372 °C, the main endothermal decomposition peak was found. These findings agreed well with the result of Sousa et al. [49]. The results show that, for some reason, the DSC decomposition peak was shifted in comparison to DTG at about 5 °C (Figure 5a,b). Nevertheless, these findings explain that torrefaction could not significantly change the properties of PLA, as the temperature was too low for efficient devolatilization.

For PAP, three peaks were observed by DTG. First at 80 °C, second at 326 °C, and third at 550 °C with 1.3%, 74.6%, and 5.4% mass change (Figure 5a, grey curve), respectively. The first and third peaks are almost not visible in Figure 5a. The first peak is related to residual water evaporation, while the second peak is probably related to cellulose decomposition. This is due to the fact that white paper is made mainly from cellulose, (85–99%) with the addition of lignin of 0–15% [50]. Nevertheless, reprocessed paper (e.g., newspaper) has less cellulose (40–55%), more lignin (18–30%), and comparable content hemicellulose (25–40%) in comparison to white paper [50]. Additionally, the previously mentioned substances could affect the PAP sample decomposition. Typically, the hemicellulose, cellulose, and lignin decompose at 225–325 °C, 305–375 °C, and 250–500 °C, respectively [51]. According to Porshnov et al. [52], the temperature range of 250–300 °C is a characteristic interval for hemicellulose decomposition, 300–350 °C for cellulose decomposition, while above 400 °C the residue of lignocellulosic substances decomposed at a very slow rate. Lignin decomposition reactions were reported to occur at up to 900 °C [52]. Therefore, it is highly probable that PAP’s third peak is related to lignin decomposition. The DSC results showed that, during PAP pyrolysis, four endothermal transformations occurred. The first transformation at 91.4 °C was probably related to residual moisture removal [53], and the following transformations were related to the decomposition of elements of the PAP sample. Similar results were obtained by Yang et al. [53], who tested clean cellulose and found the main endothermal peak related to decomposition at 355 °C. In this study, this peak was found at 329.6 °C (Figure 5b) and, similarly to the PLA, the DSC peak of PAP was shifted in comparison to DTG at about 3.6 °C. 

The kinetic parameters were determined at β = 10 °C·min^−1^ using the Coats–Redfern method. The kinetic triplets were determined for the whole process (30–800 °C) and the main peaks observed at TG/DTG plots (Figure 5a). The whole decomposition process for PAP and PLA were described by a reaction order of 1.56 and 2.02, respectively, and relative low activation energy of 33.11 kJ·(mol·K)^−1^, and 46.24 kJ·(mol·K)^−1^, respectively (Table 2). Here, it is worth noting that, for PLA, the determination coefficient was low, at 0.66, which was a result of the one-stage decomposition process, which occurred at 290–400 °C. Additionally, other kinetic triplets were determined with high determination coefficients (Table 2). The main PLA decomposition reaction was described by a reaction order of 0.42 and 160.05 kJ·(mol·K)^−1^ activation energy, while PAP exhibited an order of 2.12, and 122.55 kJ·(mol·K)^−1^ (Table 2). The first peak for PAP was omitted, as it was only residual water evaporation. It is worth noting that the suspected lignin decomposition at the third peak of the PAP sample had the highest activation energy of 173.05 kJ·(mol·K^−1^), which was about 51 kJ·(mol·K)^−1^ larger than the main decomposition of cellulose. This finding is contrary to Noszczyk et al. [54] who studied several types of biomass materials and noticed that the cellulose content had a significant impact on the E_a_, and the highest E_a_ was observed at the second stage of reaction, which was related to the cellulose decomposition [54].

### 3.4. Theoretical Mass and Energy Balance of the Torrefaction Process

Table 3 summarizes the theoretical mass and energy balance to produce 1 g CSF s given. The table compares the temperature and time. The third and fourth headings present the input mass needed to produce 1 g of CSF, and the chemical energy contained in this material. The fifth heading presents external heat provided to the torrefaction process. The sixth heading shows energy contained in 1 g of CSF. The seventh heading present a mass of gas released from the substrate during torrefaction, and the last heading show energy contained in this gas. The energy in gas was calculated as a sum of external energy provided to conduct a process and energy of substrate that was not converted into CSF.

The result shows that more PAP than PLA substrate is needed to produce 1 g of CSF. In the case of 300 °C at 60 min, the double mass of PAP is needed compared to PLA (Table 3). The reason for this large input substrate demand originates from the low mass yield of PAP torrefaction (Figure 2b). As a result, much more chemical energy contained in PAP is put into the process to produce 1 g of CSF (23,833 J for PLA vs. 43,551 J for PAP). Additionally, the DSC results showed that more energy was needed to heat PAP than PLA to 300 °C, (458 J·g^−1^_CSF_ vs. 940 J·g^−1^_CSF_) (Table 3). This is caused probably by the mostly higher specific heat value (Sp) of PAP in comparison to PLA. Depending on chemical composition, Sp of PAP varies from 1150 to 1650 J·(g·K)^−1^ [55] while, for PLA, the value varies from 1180 to 1210 J·(g·K)^−1^ [56]. On the other hand, PLA has a higher thermal conductivity, 0.12–0.15 W·(m·K)^−1^ than PAP 0.08–0.11 W·(m·K)^−1^ [55,56].

During torrefaction, torrgas are produced. The analysis showed that a small mass of torrgas is produced from PLA and, depending on process conditions, these vary from 0.004 g·g^−1^_CSF_ to 0.227 g·g^−1^_CSF_. As the production of 1 g of CSF from PAP needs far more substrate, much more torrgas is produced and varies from 0.054 g·g^−1^_CSF_ to 1.485 g·g^−1^_CSF_ (Table 3). As a result, during torrefaction at 300 °C, for each gram of produced CSF, around 1.5 g of torrgas is generated, and these torrgas contain more energy than produced CSF, while for PLA it is only 0.23 g of torrgas with around four times less energy than produced CSF (Table 3).

When energy contained in torrgas is higher than the external energy needed to produce CSF, it theoretically can be assumed that the process is self-sufficient. This is true when torrgas are incinerated to provide heat for a substrate. With that assumption can be stated that PLA torrefaction can be self-sufficient at process temperatures higher than 300 °C and 40 min, while PAP is similar from 200 °C and 20 min (Table 3). Nevertheless, these results do not include heat losses, process efficiency, and energy needed for water evaporation that is in the real feedstock. Due to many different approaches to reactors design, it is hard to assume any heat losses and process efficiency. However, the contribution of water can be calculated and added to the results obtained in this study. To remove 1% of the water from solid fuel, at least 22.57 J (2257 J·g^−1^_H2O_ is the latent heat of water evaporation at 100 °C) is needed, as well as the energy needed to heat this water to 100 °C [57]. For this reason, the herein presented calculations serve as a starting point that has to be adapted for a particular reactor system and different feedstocks.

## 4. Summary

The results of this study showed that PLA’s fuel properties cannot be improved by torrefaction, as no calorific values increase were observed with increasing process temperature and time. The reason is that PLA hardly decomposes, with negligible charring effects at torrefaction temperatures. On the other hand, PAP’s fuel properties can be improved up to 10% by applying temperatures higher than 280 °C, which is probably caused by a partial cellulose decomposition. Additionally, the kinetic analysis revealed that PLA is decomposed in a one-stage process, that takes place at ~290–400 °C, with Ea of 160.05 kJ·(mol·K)^−1^, while PAP is decomposed in a two-stage process, at ~240–400 °C, and ~668–760 °C, with Ea of 122.55 kJ·(mol·K)^−1^ and 173.05 kJ·(mol·K)^−1^, respectively. Moreover, the calculations showed that PLA torrefaction cannot be self-sufficient for CSF production and external energy is required, while CSF production from PAP proves to be self-sufficient under assumptions of no heat loss. 

These results provide the first step towards an understanding of the PLA torrefaction process, but further research is needed to investigate higher temperatures of thermal PLA processing embracing gaseous and liquid products rather than solids, as PLA decomposes entirely into volatile components. Moreover, future studies should focus on PLA co-pyrolysis with conventional plastic, as a separation in waste management facilities is currently not possible from the MSW stream. Such a separation may be possible for separately collected and clean plastic wastes, but will fail in the case of plastics with organic adhesions, which are typical for plastic in MSW. 

Regarding waste management scenarios, our study showed that the thermal properties of PLA qualify this material neither as a fuel surrogate in waste incinerators nor for an improvement by torrefaction process when we compare PLA with conventional high energy plastics. Therefore, a successive substitution of high caloric plastics by PLA may be reasonable when the end-of-life-scenario for the material is composting, but will raise the demand of conventional fuel when its thermally treated. 

## Figures and Tables

**Figure 1 materials-14-07051-f001:**
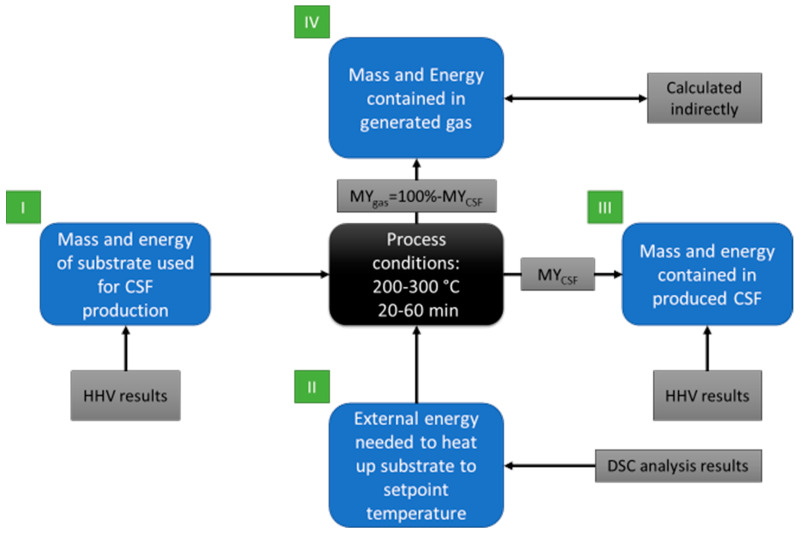
Scheme of mass and energy balance determination.

**Figure 2 materials-14-07051-f002:**
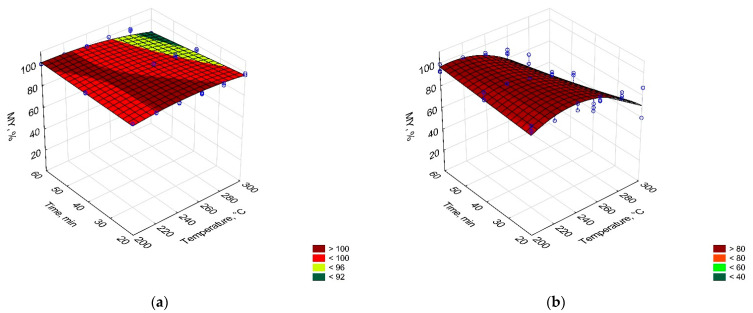
Temperature and time effect on the mass yield (MY) of carbonized solid fuel made from (**a**) PLA, (**b**) PAP.

**Figure 3 materials-14-07051-f003:**
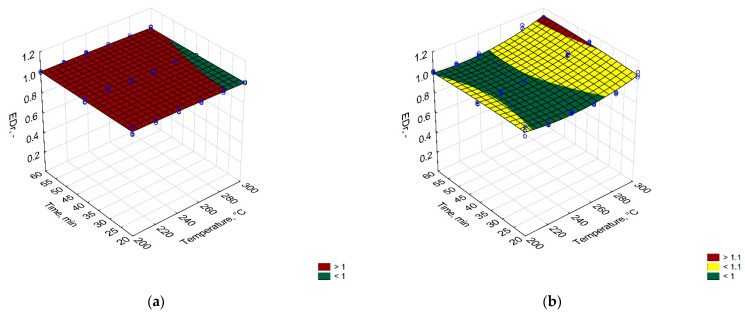
Temperature and time effect on the energy densification ratio (EDr) of carbonized solid fuel made from (**a**) PLA and (**b**) PAP.

**Figure 4 materials-14-07051-f004:**
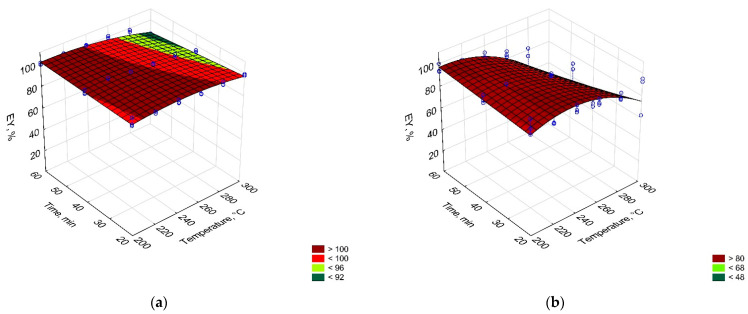
Temperature and time effect on the energy yield (EY) of carbonized solid fuel made from (**a**) PLA and (**b**) PAP.

**Figure 5 materials-14-07051-f005:**
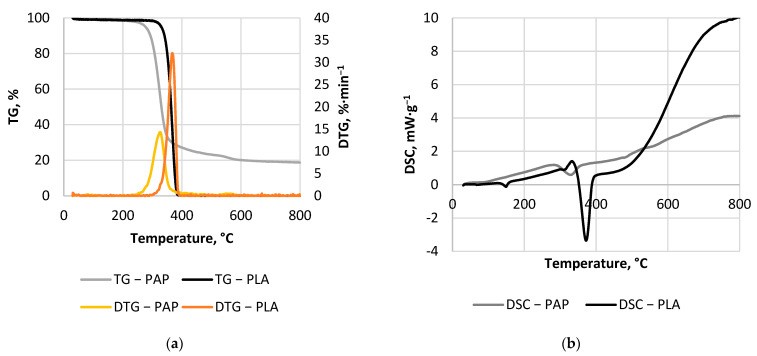
Thermal analysis results, (**a**) TG/DTG, (**b**) DSC.

**Table 1 materials-14-07051-t001:** Results of proximate analysis and calorific value of CSF, as dry basis.

Material	Temp., °C	Time, min	VM, %	FC, %	AC, %	VS, %	CP, %	HHV, J·g^−^^1^
PLA	-	-	100.0	0.0	0.0	100.0	100.0	19,420
	200	20	100.0	0.0	0.0	100.0	100.0	19,675
		40	100.0	0.0	0.0	100.0	100.0	19,598
		60	100.0	0.0	0.0	100.0	100.0	19,512
	220	20	100.0	0.0	0.0	100.0	100.0	19,631
		40	100.0	0.0	0.0	100.0	100.0	19,799
		60	100.0	0.0	0.0	100.0	100.0	19,613
	240	20	100.0	0.0	0.0	100.0	100.0	19,703
		40	100.0	0.0	0.0	100.0	100.0	19,654
		60	100.0	0.0	0.0	100.0	100.0	19,682
	260	20	100.0	0.0	0.0	100.0	100.0	19,399
		40	100.0	0.0	0.0	100.0	100.0	19,372
		60	100.0	0.0	0.0	100.0	100.0	19,592
	280	20	100.0	0.0	0.0	100.0	100.0	19,529
		40	100.0	0.0	0.0	100.0	100.0	19,510
		60	100.0	0.0	0.0	100.0	100.0	19,410
	300	20	100.0	0.0	0.0	100.0	100.0	19,346
		40	100.0	0.0	0.0	100.0	100.0	19,294
		60	100.0	0.0	0.0	100.0	100.0	19,571
PAP	-	-	88.2	8.2	3.6	96.3	96.4	17,525
	200	20	86.6	9.9	3.4	96.6	96.6	17,889
		40	86.2	10.1	3.6	96.3	96.4	17,283
		60	86.7	9.8	3.5	96.5	96.5	17,653
	220	20	88.0	8.6	3.4	96.5	96.6	17,185
		40	86.7	10.0	3.3	96.4	96.7	17,504
		60	86.4	10.1	3.5	96.4	96.5	17,368
	240	20	85.5	10.9	3.5	96.3	96.5	17,446
		40	84.7	11.8	3.6	96.2	96.4	17,366
		60	84.8	11.7	3.5	96.2	96.5	17,434
	260	20	86.2	10.2	3.6	96.1	96.4	17,163
		40	84.0	12.4	3.6	96.0	96.4	17,389
		60	81.9	14.1	4.0	95.7	96.0	17,220
	280	20	83.6	12.7	3.7	96.3	96.3	17,352
		40	67.9	26.0	6.1	93.7	93.9	19,048
		60	66.9	26.2	7.0	92.8	93.0	19,146
	300	20	69.3	24.8	5.9	93.9	94.1	18,758
		40	60.8	31.5	7.7	91.8	92.3	19,520
		60	55.7	34.6	9.7	89.9	90.3	19,346

**Table 2 materials-14-07051-t002:** Kinetic triplets determined at β = 10 °C·min^−1^ using Coats-Redfern method.

Material	Note	Temperature, °C	n	Ea, kJ·(mol·K)^−1^	A, s^−1^	R^2^
PLA	Whole process	30–800	2.02	46.24	2.91 × 10	0.66
	Main decomposition peak	290–400	0.42	160.05	2.37 × 10^10^	0.96
PAP	Whole process	30–800	1.56	33.11	5.88 × 10^−1^	0.89
	Main decomposition peak	240–400	2.12	122.55	1.74 × 10^8^	0.96
	Third decomposition peak	668–760	3.00	173.05	4.90 × 10^10^	0.91

**Table 3 materials-14-07051-t003:** Torrefaction mass and energy balance for production of 1 g of CSF from PLA and Paper wastes.

Temp., °C	Time, min	Mass of Substrate Used to Produce 1 g of CSF, g		Energy Contained in the Raw Material Used to Produce 1 g of CSF, J		External Energy Needed to Produce 1 g of CSF, J *		Energy Contained in 1 g of CSF, J **		Mass of Gas Generated during the Production of 1 g of CSF, g		Energy Contained in Gas after Production of 1 g of CSF, J ***	
		PLA	PAP	PLA	PAP	PLA	PAP	PLA	PAP	PLA	PAP	PLA	PAP
200	20	1.004	1.054	19,500	18,475	86	328	19,675	17,889	0.004	0.054	−89	914
	40	1.006	1.048	19,540	18,367	86	328	19,598	17,283	0.006	0.048	27	1412
	60	1.006	1.055	19,538	18,482	86	328	19,512	17,653	0.006	0.055	112	1157
220	20	1.003	1.074	19,483	18,817	133	425	19,631	17,185	0.003	0.074	−15	2056
	40	1.004	1.053	19,505	18,459	133	425	19,799	17,504	0.004	0.053	−161	1380
	60	1.007	1.060	19,552	18,582	133	425	19,613	17,368	0.007	0.060	72	1639
240	20	1.005	1.053	19,512	18,454	194	536	19,703	17,446	0.005	0.053	3	1543
	40	1.007	1.078	19,562	18,886	194	536	19,654	17,366	0.007	0.078	101	2056
	60	1.013	1.096	19,676	19,207	194	536	19,682	17,434	0.013	0.096	188	2309
260	20	1.010	1.066	19,608	18,683	267	663	19,399	17,163	0.010	0.066	477	2184
	40	1.011	1.102	19,642	19,308	267	663	19,372	17,389	0.011	0.102	537	2583
	60	1.007	1.170	19,562	20,499	267	663	19,592	17,220	0.007	0.170	237	3942
280	20	1.014	1.131	19,685	19,822	355	803	19,529	17,352	0.014	0.131	510	3273
	40	1.025	1.357	19,909	23,778	355	803	19,510	19,048	0.025	0.357	754	5534
	60	1.022	1.550	19,839	27,163	355	803	19,410	19,146	0.022	0.550	784	8820
300	20	1.012	1.288	19,646	22,571	458	940	19,346	18,758	0.012	0.288	758	4753
	40	1.043	2.357	20,247	41,303	458	940	19,294	19,520	0.043	1.357	1,410	22,722
	60	1.227	2.485	23,833	43,551	458	940	19,571	19,346	0.227	1.485	4,719	25,144

* value determined using DSC analysis result. ** value determined using calorimetric analysis result (HHV). *** value is the sum of chemical energy contained in gas and heat from external energy, assuming that no external energy stays in CSF.

## Data Availability

All data derived during the experiments are given in the paper or the Appendix A.

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
