# Peer review of "Carbonized Solid Fuel Production from Polylactic Acid and Paper Waste Due to Torrefaction"

_materials, 2021, doi:10.3390/ma14227051_

Round 1
Reviewer 1 Report
I think it's a good job, but it needs to correct different aspects. They are the following:
There are many abbreviations, so a Nomenclature section is recommended.
It would be advisable to complete the introduction on TG/DTG/DSC thermal analysis and that these are commonly performed studies, citing references for other waste would be recommendable, for example:
Control of several emissions during olive pomace thermal degradation
International Journal of Molecular SciencesOpen AccessVolume 15, Issue 10, 13 October 2014, Pages 18349-18361
It would be interesting to incorporate some analysis on emissions, similar to the previous paper.
It is also necessary to correct the following errors:
Line 90, a.k.a. Not recommended for use.
Line 120, Insert a comma.
It is recommended to complete the definition of materials.
Line 275, Insert a comma.
It is recommended to reduce font size in Tables 1, 2 and 3.
Thank you so much.
Best regards.
Reviewer 2 Report
The manuscript discusses an industrially relevant topic for waste to value generation and using polylactic acid and paper waste to develop carbonized solid fuel through torrefaction process. The article is well written and explains results in-line with the observations. However, there are few changes which are required prior to the acceptance of the manuscript for publication.
The suggestions are as follows:
- Please include the standard deviation in the data, especially for the thermal analysis and calorific value. The current accuracy for the residual volume and volatile is under doubt and may not be completely reproducible.
- Please summarize the conclusions. The current version is very much like an outlook of a review for torrefaction technology.
- Please increase the size of text in Figure 2, Figure 3 and Figure 4 for ease of reading. The current text is too small.
- The regression coefficient is quite low for the kinetic data fit and therefore the applicability of the Coats-Redfern (CR) method is questionable. Because it is known that for reactions with mixed mechanism, the CR method exhibits nonlinear trends. And therefore, the reaction models and kinetic parameters can not be extracted. Kindly explain why does authors selects this method despite of low R2 value?
